# Membership Inference Attacks
# *via* Adversarial Examples

**Hamid Jalalzai**[*]
Laboratoire d'Informatique de l'École polytechnique
Inria, Palaiseau, France
`hamid.jalalzai@inria.fr`

**Elie Kadoche**[*]
LTCI, Télécom Paris
Institut Polytechnique de Paris, France
`elie.kadoche@telecom-paris.fr`

**Rémi Leluc**[*]
LTCI, Télécom Paris
Institut Polytechnique de Paris, France
`remi.leluc@telecom-paris.fr`

**Vincent Plassier**[*]
CMAP, Ecole Polytechique
Lagrange Research Center, Paris, France
`vincent.plassier@polytechnique.edu`

## Abstract

The rise of machine learning and deep learning has led to significant improvements in several areas. This change is supported by both the dramatic increase in computing power and the collection of large datasets. Such massive datasets often contain personal data that can pose a threat to privacy. Membership inference attacks are an emerging research direction that aims to recover training data used by a learning algorithm. In this work, we develop a means to measure the leakage of training data leveraging a quantity appearing as a proxy of the total variation of a trained model near its training samples. We extend our work by providing a novel defense mechanism. Our contributions are supported by empirical evidence through convincing numerical experiments.

## 1 Introduction

At the forefront of artificial intelligence (AI), the benefits and outstanding performance of modern machine learning cannot be overlooked. Through the lens of supervised learning, computers can now compete with humans regarding vision [28, 30, 38, 61], object recognition [16] or medical image segmentation [39]. Unsupervised and semi-supervised learning techniques led to an impressive development of natural language processing [18, 36, 37, 76] or language understanding [12, 24, 81, 84]. More recently, reinforcement learning [70] has reached superhuman levels in a wide range of tasks such as games [54, 67], complex biological tasks [40] or even smart devices and smart cities [55]. Most of the aforementioned methods use Deep Learning [43] to achieve peak performance. However, these methods usually require a large amount of data to achieve generalization [2, 22, 45, 75] and nowadays large data sets used in the era of so-called *big data* [83] can be easily collected.

Information Disclosure [44] describes the involuntary leakage of information from a provider (*e.g.* a website database, an API or even a Deep Learning model). Machine learning models used in production do not necessarily provide fairness [53, 78], security [4, 11] or privacy considerations [35]. Before claiming that leakage of sensitive information is *not a bug, but a feature* [26], privacy protection of data providers to machine learning models has become necessary. Indeed, it is required under recent data regulation (see CCP [1], Villani et al. [77] and Regulation [62], van der Burg et al. [73]). It is a growing direction of research [17, 52].

Recent contributions from the fields of deep learning [6, 10, 63] and privacy and security [21, 47, 64]

---

[*]Equal contribution.

2022 Trustworthy and Socially Responsible Machine Learning (TSRML 2022) co-located with NeurIPS 2022.

show that neural networks and deep models are likely to behave like interpolators in high dimensions. Considered benign for generalization [9, 49, 72], overfitting and thus overconfident wrong predictions can be harmful [4, 64] and uncalibrated [60, 79].

Sharing data and model in a private way is a major barrier to the adoption of new AI technologies [58]. Therefore, ensuring that data-driven technologies respect important privacy constraints and government regulations is at the core of modern AI security [25]. In this way, this paper attempts to measure the extent of information disclosure or data leakage that a malicious attacker exploiting the vulnerabilities of a given model can potentially recover. Defense mechanisms that prevent or minimize information leakage are desirable tools to ensure the confidentiality of training data. We propose a defense mechanism that prevents the leakage of training data with membership inference attacks based on adversarial strategies.

**Contributions.** *(i)* We describe a novel framework for membership inference attacks (MIA) using adversarial examples with general functionals in the output space; *(ii)* We present a particular method based on the total variation of the score function; *(iii)* We propose a defense mechanism against MIA leveraging adversarial examples deferred to the supplementary material.

**Outline.** Section 2 introduces the relevant background for the analysis. Section 3 provides our novel MIA framework using counterexamples in the output space. Section 4 gathers numerical experiments. Finally, Section 5 concludes our work with a discussion of further avenue. The proofs and relevant material are given in the Appendix.

## 2 Framework

**Statistical Learning & Pattern Recognition.** Statistical learning [29] is a branch of machine learning that deals with the problem of statistical inference, i.e., constructing a prediction function from data. In pattern recognition [14], a discrete random label $Y$ valued in $\mathcal{Y}$, with cardinality arbitrarily set to $K > 1$, is to be predicted based on the observation of a random vector $X$ valued in an input space $\mathcal{X}$ using the classification rule minimizing a loss function. To perform pattern recognition, the empirical risk minimization (ERM) learning paradigm [74] suggests to solve

$$g^\star \in \arg\min_{g \in \mathcal{G}} \mathbb{E}\left\{\ell(g(X), Y)\right\},$$

where the minimization is done over a class of classifiers $\mathcal{G}$ and the risk $\mathbb{E}\left\{\ell(g(X), Y)\right\}$ is associated with a loss $\ell$ measuring the error between the prediction $g(X)$ and the label $Y$. Since the joint distribution $P_{X,Y}$ is generally unknown, one may rely on a training dataset $\mathcal{D}_{\text{train}} = \{(X_i, Y_i)\}_{i=1}^n$ consisting of $n \geq 1$ i.i.d copies of $(X, Y)$ to minimize the empirical risk

$$\widehat{g} \in \arg\min_{g \in \mathcal{G}} \left\{\widehat{R}(g) = \frac{1}{n}\sum_{i=1}^n \ell(g(X_i), Y_i)\right\}.$$

In the case of deep neural networks, the empirical risk minimization can be rewritten as minimization over a set $\Theta$ corresponding to the set of weights associated with a given neural network architecture $\mathcal{G}$. In full generality, the minimization should occur over $\mathcal{G} \times \Theta$, although for the sake of simplicity, it is assumed that the deep models'architecture is fixed and well-designed. A classifier $g_\theta \in \mathcal{G} \times \Theta$ yields a score distribution on the label set $\mathcal{P}(\mathcal{Y})$ that hopefully approximates the probability distribution $\mathbb{P}\left(Y | X = x\right)$ [33]. Consistently with our notation, let $\widehat{g}_\theta$ denote a minimizer over $\Theta$ of the risk $\widehat{R}(g_\theta)$.

**Membership Inference Attack.** To measure the leakage of information of a model, the authors of [65] introduced the paradigm of Membership Inference Attacks (MIA). The goal of MIA is to determine whether a given labeled sample $(x, y) \in \mathcal{X} \times \mathcal{Y}$ belongs to the set of training data $\mathcal{D}_{\text{train}}$ of a classifier $\widehat{g}_\theta$. Without loss of generality, most MIA strategies determine whether a sample was used to train a particular model based on decision rules involving indicators functions. Therefore, the MIA strategies can be written as follows

$$x \mapsto \mathbb{1}\{s(x) \geq \tau\}$$

where $s : \mathcal{X} \mapsto \mathbb{R}$ is a score function and $\tau \in \mathbb{R}$ is a threshold. The score function $s$ may rely on the loss value [65], the gradient norm [66, 82], or on a modified entropy measure [68, 69].

**Adversarial Examples.** Evasion attacks [7, 8, 13, 51] of classifiers are referred to adversarial attacks when dealing with neural networks [13]. As a tool that contributes to a better understanding of the underlying estimators, adversarial strategies are attracting increasing interest in various areas of machine learning such as computer vision [3], reinforcement learning [42], and ranking data [31]. Two types of adversarial strategies can be considered: *targeted attacks* and *untargeted attacks*.

For a given sample $x \in \mathcal{X}$ labeled by a classifier $\widehat{g}_\theta$ as $y \in \mathcal{Y}$, a targeted attacker $\psi$ with target label $\widetilde{y} \neq y$ builds a carefully crafted noise $\varepsilon \in \mathbb{R}^d$ such that $x + \varepsilon$ is predicted as $\widetilde{y}$. On the other hand, regarding untargeted attacks, an attacker $\psi$ will generate a noise $\varepsilon \in \mathbb{R}^d$ for a given sample $x \in \mathcal{X}$ such that: $\arg\max \widehat{g}_\theta(x) \neq \arg\max \widehat{g}_\theta(x + \varepsilon)$. Additional constraints on the noise $\varepsilon$ are common to provide the smallest perturbation that gives the desired results. In the numerical experiments of Section 4, adversarial examples are built using the FAB adversarial strategy [19] from Torchattacks library [41]. One such scheme aims to provide the smallest perturbation $\varepsilon$ to fool the target model $\widehat{g}$ (see Figure 5 in Appendix C.1)

**Learning with noisy labels.** The authors of [56] introduced the formal framework of learning with noisy labels. They studied the binary classification problem in the presence of random classification noise and provided a surrogate objective. Instead of seeing the true label, a learner sees a sample with a label that has been flipped independently. To the best of our knowledge, there is no paper that relies on injecting voluntary noise into the labels to prevent deep models leakage with peaks of confidence on training data samples.

## 3 MIA Methodology

This section provides insights about the methodology behind MIA with adversarial examples. First, we motivate the strategy by using synthetic data to show that trained classifiers exhibit local *peaks of confidence* around the training samples. These peaks are then exploited to perform MIA using general functionals in the output space.

### 3.1 Motivations

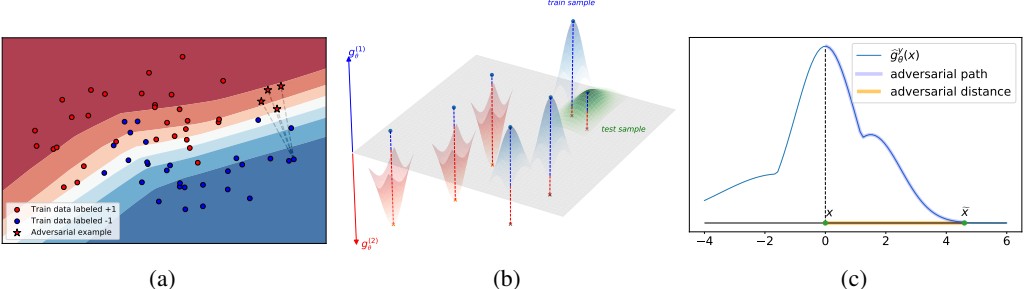

Figure 1: (1a) Illustration of adversarial examples (red stars) resulting from train data (blue circle). The background color represents the probability output of a binary classifier. (1b) Illustration of local confidence peaks on the landscape of a binary classifier. The vertical axis in blue (*resp.* red) represents the score for class 1 (*resp.* 2). (1c) Illustrative MIA measures (adversarial path and adversarial distance) with a classifier $\widehat{g}_\theta$ where $\widehat{g}_\theta^y(x)$ denotes $\widehat{g}_\theta$'s score for the class $y$ of a univariate sample $x$ labeled $y$. $\widetilde{x}$ represents an adversarial example for the sample $x$ regarding $\widehat{g}_\theta$.

**Excess of confidence.** The starting point of the method is based on the following conjecture: *"trained classifiers are over-confident on their training datasets because they exhibit local peaks of confidence around the training samples"*. More precisely, we argue that the shape of the landscape of a trained binary classifier consists of high hills and valleys around the training samples and flatter curves for new, unseen data. While Figure 1a depicts the landscape of a binary classifier in a cut-away view, its general shape takes the form described in Figure 1b, where the scores are sharp around samples from the train set and flat around the new unseen data.

To highlight such behavior with *peaks of confidence*, we place ourselves in the classification framework with synthethic data. The goal here is to assess the following statement: for a trained classifier, a small amount of noise is sufficient to change the distribution of the score function on the

training data, whereas a larger amount of noise is required to change the distribution of the score function on new unseen data. Intuitively, more noise should be required near a training sample to fool a classifier, as the sample must move farther away from the local excess of confidence.

**Synthetic framework.** Consider the multiclass classification task with a dataset $\mathcal{D}$ consisting of $n \geq 1$ observations $X_1, \ldots, X_n \in \mathbb{R}^d$ along with their associated labels $y_1, \ldots, y_n \in \{1, \ldots, K\}$. The data is split into a training set $\mathcal{D}_{\text{train}}$ and a testing set $\mathcal{D}_{\text{test}}$. For ease of notation, denote by $\mathcal{D}_a$ with $a \in \{\text{train}, \text{test}\}$ the different datasets. For each class $k \in \{1, \ldots, K\}$, denote by $\mathcal{D}_a(k) = \{(X, y) \in \mathcal{D} : X \in \mathcal{D}_a, y = k\}$ the elements of $\mathcal{D}_a$ which are labeled $k$ and $n_a(k) = \sum_{(X,y) \in \mathcal{D}_a} \mathbb{1}\{y = k\}$ its cardinal number. For $k \in \{1, \ldots, K\}$, denote by $\bar{X}_a(k) = \sum_{X \in \mathcal{D}_a(k)} X/n_a(k)$ the barycenter of class $k$ and consider the average pairwise intra-cluster distance $\delta_a(k)$ defined by $\delta_a(k) = \sum_{i,j=1}^{n_a(k)} \|X_i - X_j\|_2 / \left(2(n_a(k) - 1)n_a(k)\right)$.

**Adversarial examples.** Each sample $X$ is gradually modified by a random noise $\xi(t)$ and the distribution of the scores $\rho_X : t \mapsto \|\hat{g}_\theta(X + \xi(t))\|_\infty$ are to be compared. The random noise is a Gaussian anisotropic noise directed towards the closest barycenter of $X$ that is differently labeled. More precisely, given a sample $X \in \mathcal{D}_a(k)$, consider the barycenter $\bar{X}_a(l)$ which is the closest to $X$ and has a different label, *i.e.*, $l = \arg\min_{j \neq k} \|X - \bar{X}_a(j)\|_2$. The underlying unit direction is $v = (X - \bar{X}_a(l))/\|X - \bar{X}_a(l)\|_2$ and the sample $X$ is modified as

$$X_t = X + \xi(t), \quad \xi(t) = |\varepsilon_t|.v$$

where $\varepsilon_t \sim \mathcal{N}(0, t\delta_a(k))$ controls the level of noise required to disturb the sample. For each value of $t$, denote by $\mathcal{D}_{\text{train}}^t$ and $\mathcal{D}_{\text{test}}^t$ the corresponding datasets consisting of the modified samples $X_t$.

For different values of $t$, univariate Kolmogorov-Smirnov (KS) tests are performed to compare the distributions of the scores on the initial data $\rho_X(0) = \|\hat{g}(X)\|_\infty$ and their noisy counterparts $\rho_X(t) = \|\hat{g}_\theta(X + \xi(t))\|_\infty$ for both the train and test sets. Furthermore, we also perform bivariate two sample KS tests (see [27]) directly on the data from the same pairs $\left(\mathcal{D}_{\text{train}}, \mathcal{D}_{\text{train}}^t\right)$ and $\left(\mathcal{D}_{\text{test}}, \mathcal{D}_{\text{test}}^t\right)$ to ensure that we do not disturb the data too much. The $p$-values are averaged over the total number of classes in their respective pairs. It is straightforward that as the value of $t$ increases, the classifier's score on the initial data and the score on the noisy data are getting increasingly different, hence the $p$-value of the involved tests decreases. Although, in the idea that trained classifiers present local confidence peaks, it is expected that the $p$-values for the KS test on $(\rho_X(0), \rho_X(t))$ for $\mathcal{D}_{\text{train}}$ decrease faster than the respective ones for $\mathcal{D}_{\text{test}}$.

**Results.** Figure 2 (Left) shows the evolution of $p$-values from the KS tests performed on $\rho_X(0)$ and $\rho_X(t)$ where the number of classes equals 4. As expected, the distributional gap between the original training data $\mathcal{D}_{\text{train}}$ and its noisy counterpart $\mathcal{D}_{\text{train}}^t$ is more remarkable, the more noise is added compared to the test data $\mathcal{D}_{\text{test}}$ and its noisy counterpart $\mathcal{D}_{\text{test}}^t$, as suggested by the lower median $p$-values. The boxplots in Figure 2 (Right) provide the evolution of the $p$-values from a bivariate KS test directly computed on the input data to ensure that the procedure does not excessively disrupt the data. Such a test describes the evolution of the distributional influence of the noise level $t$. Note that the decay rate of $p$-values with $t$ is similar between $(\mathcal{D}_{\text{train}}, \mathcal{D}_{\text{train}}^t)$ and $(\mathcal{D}_{\text{test}}, \mathcal{D}_{\text{test}}^t)$. This behavior along with the left-hand boxplots confirm that the changes in the output space do not depend on the perturbations made in the input data, but rather on the confidence peaks in the training data.

As a consequence of this first motivational experiment with some *simplistic* noise as adversarial strategy, the local excess of confidence of a classifier transcribes the underlying presence of training data. As a result, the output manifold of the classifier can be scrutinized to perform membership inference attacks..

## 3.2 Arc length & Total Variation

In order to measure what happens on the manifold of predicted values, we consider curve lengths along adversarial paths. For any real-valued continuous function $f$, the total variation is a measure of the one-dimensional arc length of the curve with parametric equation $t \mapsto f(t)$.

**Definition 1 (Total variation for univariate function $\mathbb{R}$)** *Consider a real-valued function $f$ defined on an interval $[a, b] \subset \mathbb{R}$ and let $\Pi$ denote the set of all partitions i.e. $\Pi = \{\pi = $*

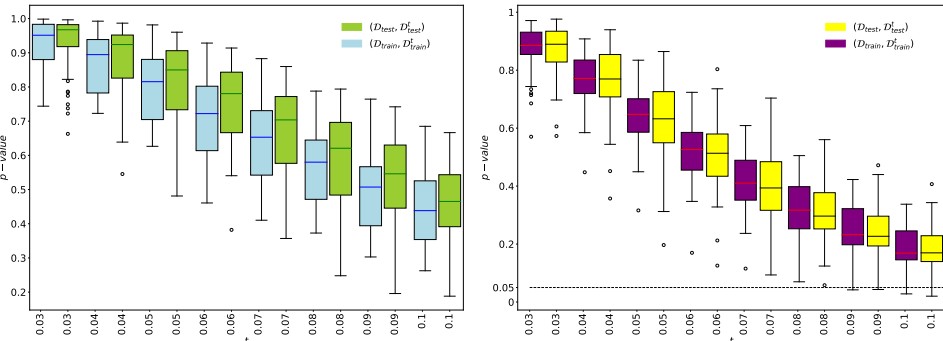

Figure 2: (Left) Boxplot of univariate two sample Kolmogorov-Smirnov tests average $p$-values between max scores provided by $\widehat{g}_\theta$ computed on the original train data $\mathcal{D}_{\text{train}}$, (*respectively* test data $\mathcal{D}_{\text{test}}$) and the noisy train counterpart $\mathcal{D}_{\text{train}}^t$ (*respectively* noisy test counterpart $\mathcal{D}_{\text{test}}^t$) with varying noise level $t$. (Right) Boxplot of bivariate two sample Kolmogorov-Smirnov tests average $p$-values between the original train data $\mathcal{D}_{\text{train}}$ (*respectively* test data $\mathcal{D}_{\text{test}}$) and the noisy train counterpart $\mathcal{D}_{\text{train}}^t$ (*respectively* noisy test counterpart $\mathcal{D}_{\text{test}}^t$) with varying noise level $t$. The boxplots display the results for 50 different (random) iterations of the experiment. The number of classes is equal to 4.

$(x_0, \ldots, x_n)$ s.t $a = x_0 < \ldots < x_n = b\}$ *of the given interval. The total variation of $f$ on* $[a,b]$ *is given by* $V_{[a,b]}(f) = \sup_{\pi \in \Pi} \sum_{k=1}^{n} |f(x_k) - f(x_{k-1})|$. *If $f$ is differentiable, we have* $V_{[a,b]}(f) = \int_a^b |f'(t)|\mathrm{d}t$.

**Output space functionals.** Given a classifier $g \in \mathcal{G}$ and a sample $x \in \mathcal{X}$, standard MIA strategies determine whether $x$ is part of the training set or not using $\mathbb{1}\{s(x) \geq \tau\}$ where $s : \mathcal{X} \mapsto \mathbb{R}$ is a score function. This scheme can be extended to account for the information brought in by adversarial examples. For any sample $x \in \mathcal{X}$ with associated adversarial example $\widetilde{x} \in \mathcal{X}$, a general MIA strategy may be written as

$$x \mapsto \mathbb{1}\{\varphi(g, x, \widetilde{x}) \geq \tau\}, \tag{1}$$

where $\varphi : \mathcal{G} \times \mathcal{X} \times \mathcal{X} \to \mathbb{R}$ is a functional to be specified. When using $\ell_p$-norms in the input space as $\varphi(\widehat{g}_\theta, x, \widetilde{x}) = \|\widetilde{x} - x\|_p$, the rule of Equation (1) recovers the framework of [21]. Such distances denoted as *adversarial distances* by [21] are only concerned by the input space $\mathcal{X}$ and do not directly take into account the behavior of the classifier $\widehat{g}_\theta$ and its predicted values. Instead, we advocate for functionals $\varphi : \mathcal{G} \times \mathcal{P}(\mathcal{Y}) \times \mathcal{P}(\mathcal{Y})$ that focus on distances in the output space.

Motivated by the illustration of Figure 1b, we argue that more attention is needed on the manifold of predicted values $g(\mathcal{X})$ rather than the aforementioned *adversarial distances*. Indeed, the landscape is more likely to capture the high confidence levels associated to the samples used during the training phase. Such behavior is depicted in Figure 1c below where the arc length of the adversarial path better transcribes the local excess of confidence than the adversarial distance.

Consider $x \in \mathcal{X}$ and its associated minimal adversarial counterpart $\widetilde{x} = x + \varepsilon$. The adversarial path is the path connecting the max scores of $x$ and $\widetilde{x}$ along the manifold of predicted values. These paths are characterized by the following family of parameterizations.

**Definition 2 (Adversarial paths)** *The family of adversarial paths is parameterized by $(\gamma_x)_{x \in \mathcal{X}}$ where for all $x \in \mathcal{X}, \gamma_x : [0,1] \to [0,1]$ with $\gamma_x : t \mapsto \|\hat{g}_\theta(x + t\varepsilon)\|_\infty$.*

**Approximated total variation.** The associated arc length of an adversarial path is $\mathscr{L}(\gamma_x) = \int_0^1 |\gamma_x'(t)|\mathrm{d}t$. Using definition 2, the MIA rule we propose is given by

$$x \mapsto \mathbb{1}\{\mathscr{L}(\gamma_x) \geq \tau\}. \tag{2}$$

However, the underlying score function in Equation (2) may be intractable in practice. Hence, one needs to rely on an estimate version of such curvature length. Consider $N \in \mathbb{N} \setminus \{0\}$ and the partition sequence $(t_k)_{k<N}$ given by $0 = t_{-1} = t_0 < \ldots < t_N = 1$ where $t_k = k/N$. We approximate the arc length $\mathscr{L}(\gamma_x)$ by $\mathscr{L}_N(\gamma_x) = \sum_{k=0}^{N-1} |\gamma_x(t_k) - \gamma_x(t_{k-1})|$.

Accordingly, Algorithm 1 provides the pseudocode of our MIA strategy based on the adversarial path and its approximated curvature length.

---

**Algorithm 1** SISYPHOS

---

**Require:** Target sample $(x_{\text{test}}, y_{\text{test}})$, target model $\widehat{g}_\theta$, adversarial strategy $\psi$, $N \in \mathbb{N} \setminus \{0\}$ and $\tau \in \mathbb{R}_+$.
  1: Build $\widetilde{x}_{\text{test}} \leftarrow \psi((x_{\text{test}}, y_{\text{test}}), \widehat{g}_\theta)$.
  2: Compute $\mathscr{L}_N(\gamma_{x_{\text{test}}})$
  3: **Return** $\mathbb{1}\{\mathscr{L}_N(\gamma_{x_{\text{test}}}) \geq \tau\}$.

---

Note that with Riemann sums, we have $\mathscr{L}_N(\gamma_x) \to \mathscr{L}(\gamma_x)$ as $N \to +\infty$. More precisely we have the following upper bound whose proof is deferred to Appendix A.

**Proposition 1** *Assume $\gamma_x \in \mathcal{C}^1([0,1])$ and denote $(I_j)_{j \in J}$ the intervals where $\gamma_x$ is strictly monotone. If the subdivision $(t_k)_{k=1}^N$ satisfies $\sup_{k=1}^N (t_k - t_{k-1}) \leq \inf_{j \in J} |I_j|$, then we have*

$$0 \leq \mathscr{L}(\gamma_x) - \mathscr{L}_N(\gamma_x) \leq \|\gamma_x'\|_\infty |J| \sup_{k=1}^N (t_k - t_{k-1}).$$

**Remark 1** *If $\widehat{g}_\theta$ is Lipschitz then the framework of [21] is covered by Equation (2).*

# 4  Numerical Experiments

Hereafter we focus on numerical experiments regarding MIA. Further numerical experiments illustrating a defense mechanism are deferred to Appendix D.3.

## 4.1  Experimental Setting

**MIA Baseline.** We place ourselves in the experimental framework of [21] and compare their MIA strategy to Sisyphos (Algorithm 1) as their method is essentially the main white box comparable adversarial example-based membership inference attack. Adversarial examples are built with FAB adversarial strategy [19]. Experiments conducted in Appendix C and Appendix E illustrate and discuss broader experimental frameworks.

**Target Dataset.** The real world dataset CIFAR10 consists of $60,000$ colored images of size $32 \times 32$ associated with $10$ different labels. These data are divided into two subsets: the training set containing $50,000$ images and the $10,000$ remaining samples belonging to the test set which corresponds to the unseen data.

**Target Neural Networks.** We consider the Resnet model [34] publicly available in the following repository[2]. For the sake of comparison, we take the same weights as used in [21] to perform MIA.

## 4.2  Results

Table 1 and Figure 3 provide the comparison's result of Sisyphos and [21]. Based on our MIA experiment, Sisyphos outperforms the method of [21] regarding both AUC and accuracy.

---

[2]https://github.com/bearpaw/pytorch-classification

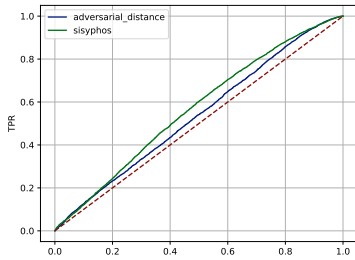

Figure 3: Comparison of ROC curves of MIA strategies relying on adversarial strategies against Resnet on CIFAR10.

| MIA Strategy | Resnet | |
|---|---|---|
| | AUC | Accuracy |
| [21] | 0.54 | 0.54 |
| Sisyphos (Ours) | **0.59** | **0.57** |

Table 1: Comparison of MIA strategies relying on adversarial examples. We report AUC (%) and Accuracy (%) scores on a balanced evaluation set. $10k$ are uniformly selected from the training set (members) and the whole $10k$ samples from the testing set are selected (non-members). All the data selected is used for evaluation.

## 5   Conclusion and Future Works

We addressed the problem of membership inference attacks based on adversarial examples relying on general functionals. Classifiers' excess of confidence on their training examples poses a privacy threat. We introduced a defense mechanism mitigating the risks of attacks using adversarial examples. Future work will explore relevant privacy preserving means with gradient flows [5, 50] to provide a better understanding of learning with privacy. The MIA risks of classifiers partitioning the input space with hyperplanes [15, 57] will be further investigated.

**Acknowledgment.** This research was supported by the ERC project Hypatia under the European Unions Horizon 2020 research and innovation program. Grant agreement No. 835294.

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

# SUPPLEMENTARY MATERIAL: MEMBERSHIP INFERENCE ATTACKS VIA ADVERSARIAL EXAMPLES

Appendix A gathers some mathematical proofs. Appendix B provides additional details concerning the synthetic framework presented in Section 3. Appendix C collects a detailed comparison of different MIA strategies, especially regarding the construction and properties of the adversarial noise built in popular adversarial attacks libraries. Appendix D presents our defense mechanism while Appendix D.3 is a numerical illustration of the defense mechanism with the label noise injection. Finally, Appendix E invites to MIA risks based on adversarial examples in a broader scope.

## A    Proofs

Appendix A gathers proofs deferred from the main paper.

### A.1    Proof of Proposition 1

PROOF. Since $\gamma'_x$ is continuous, note that $|J| < \infty$. In addition, for any $j \in J$ consider $K_j = \{k \in \{1, \dots, N\} : ]t_{k-1}, t_k[ \subset I_j\}$ and define $\bar{K} = \{1, \dots, N\} \setminus (\cup_{j \in J} K_j)$. Since $\gamma_x$ is supposed in $\mathcal{C}^1([0,1])$, using Taylor's theorem we obtain

$$\gamma_x(t_k) - \gamma_x(t_{k-1}) = \int_{t_{k-1}}^{t_k} \gamma'_x(t)\mathrm{d}t.$$

Thus, we deduce that

$$
\begin{aligned}
&\mathscr{L}_N(\gamma_x) \\
&= \sum_{k=1}^N |\int_{t_{k-1}}^{t_k} \gamma'_x(t)\mathrm{d}t| \\
&= \sum_{j \in J} \sum_{k \in K_j} |\int_{t_{k-1}}^{t_k} \gamma'_x(t)\mathrm{d}t| + \sum_{k \in \bar{K}} |\int_{t_{k-1}}^{t_k} \gamma'_x(t)\mathrm{d}t| \\
&= \mathscr{L}(\gamma_x) + \sum_{k \in \bar{K}} \left( |\int_{t_{k-1}}^{t_k} \gamma'_x(t)\mathrm{d}t| - \int_{t_{k-1}}^{t_k} |\gamma'_x(t)|\mathrm{d}t \right).
\end{aligned}
$$

Hence, we have

$$0 \le \mathscr{L}(\gamma_x) - \mathscr{L}_N(\gamma_x) \le \sum_{k \in \bar{K}} \int_{t_{k-1}}^{t_k} |\gamma'_x(t)|\mathrm{d}t. \tag{3}$$

For any $k \in \bar{K}$, either $]t_{k-1}, t_k[ \cap (\cup_{j \in J} I_j) = \emptyset$ but it yields $\int_{t_{k-1}}^{t_k} |\gamma'_x| = 0$; or there exists $j \in J$ such that $]t_{k-1}, t_k[ \cap I_j \ne \emptyset$ and therefore $t_{k-1} < \sup I_j < t_k$, but using $\sup_{k=1}^N (t_k - t_{k-1}) \le \inf_{j \in J} |I_j|$ ensures the uniqueness of $j \in J$ satisfying $]t_{k-1}, t_k[ \cap I_j \ne \emptyset$. Hence, combining (3) with the previous line gives

$$
\begin{aligned}
\sum_{k \in \bar{K}} \int_{t_{k-1}}^{t_k} |\gamma'_x(t)|\mathrm{d}t &= \sum_{j \in J} \sum_{t_{k-1} < \sup I_j < t_k} \int_{t_{k-1}}^{t_k} |\gamma'_x(t)|\mathrm{d}t \\
&\le \|\gamma'_x\|_\infty |J| \sup_{k=1}^N (t_k - t_{k-1}),
\end{aligned}
$$

which concludes the proof.

### A.2  Proof of Remark 1

PROOF. The decision rule of the proposed approach is $D(x) = \mathbb{1}\{\mathscr{L}(\gamma_x) \geq \tau\}$. Using that $\widehat{g}_\theta$ is $\lambda$-Lipschitz, it is straightforward to make the link between the input and output space as follows

$$D(x) = \mathbb{1}\{\mathscr{L}(\gamma_x) \geq \tau\}$$
$$= \mathbb{1}\{\sup_{\pi \in \Pi} \sum_{k=1}^{n} |\widehat{g}_\theta(x_k) - \widehat{g}_\theta(x_{k-1})| \geq \tau\}$$
$$= \mathbb{1}\{\lambda \|\varepsilon\| \geq \tau\}$$
$$= \mathbb{1}\{\|\varepsilon\| \geq \tau/\lambda\},$$

which recovers the decision rule of [21] of the form $x \mapsto \mathbb{1}\{\|\varepsilon\| \geq \tau'\}$ with $\tau' = \tau/\lambda$. Note that in practice and in the experiments conducted in Section 4, the values of $\tau$ can be optimized through cross validation.

## B  Further Details on Motivational Experiments

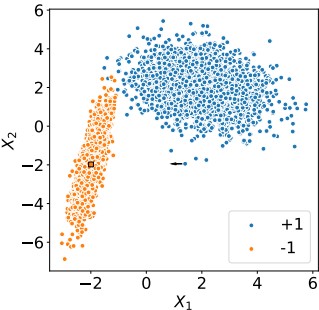

Figure 4: Illustration of the *simplistic* adversarial noise (black arrow) for a bivariate sample labeled $+1$ directed towards the barycenter of data labeled $-1$ (black square).

In Section 3, the simulated data is generated with `make_classification` function from scikit learn [59], the number of cluster per class is set to 1 as illustrated in Figure 4. The number of sample is set to 2000. The class of classifier $\mathcal{G}$ is the class of multi-layer perceptrons with 2 hidden layers of size 10. The number of iterations is set to 5000 to guarantee convergence of the classifier's weights. The boxplots are obtained over 50 iterations of the experiment.

## C  Comparing MIA based on Adversarial Strategies: an Ablation Study

In this section, we present the main difference of the MIA strategies mentioned in the paper, namely Sisyphos (Algorithm 1) and [21]. First, we recall that [21]'s MIA strategy corresponds to a specific case of functional regarding the MIA framework introduced in this paper. In a similar way as results depicted in [46], through an ablation study we analyze the influence of three elements. First, we study in Section C.1, the influence of the adversarial strategy. In Section C.2, we focus on the influence of the experimental framework. Finally in Section C.3, the influence of the classifier pre-trained model is discussed.

### C.1  Influence of the Adversarial Strategy

Hereafter the difference between two adversarial strategies are depicted. [21] leverages adversarial examples based on `AutoAttack` [20]. In Experiments from Section 4, the chosen adversarial strategy is `FAB` [19]. Let $(x_i, y_i)$ denote an image sample $x_i$ from CIFAR10 with label $y_i$. Let $\varepsilon_i$ be some adversarial noise fooling Resnet neural network on $x_i$ obtained either with `FAB` or `AutoAttack`. Figure 5 reports the results of Resnet's accuracy (%) on the partially noisy samples $x_i + \frac{k}{N}\varepsilon_i$. It appears that the partial noise $\frac{k}{N}\varepsilon_i$ is more likely to fool Resnet classifier with $\varepsilon_i$ is obtained with `AutoAttack` than `FAB`. Hence one may prefer to work with `FAB` to obtain minimal noise required to perform some adversarial strategy.

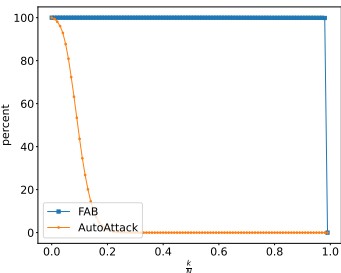

Figure 5: Evolution of the ratio (%) of correctly classified samples $x_i + \frac{k}{N}\varepsilon_i$ as a function of $\frac{k}{N}$ for increasing values of $k$ with $0 \leq k \leq N$. $\varepsilon_i$ is built either with `AutoAttack` or `FAB`.

## C.2  Influence of the Experimental Framework

To train neural network on images, padding or image rotation are common augmentation schemes [80]. Without loss of generalization, one may assume that some random input image to be classified -after the training phase- is not padded. Figure 6 shows examples of CIFAR10 images without padding and their respective counterpart with padding or rotation.

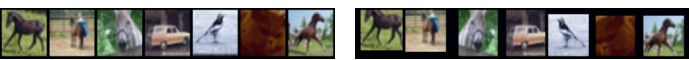

Figure 6: (Left) Input CIFAR10 images and (Right) corresponding images with padding or rotation.

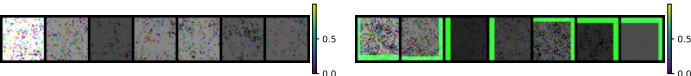

Figure 7: (Left) `FAB` Adversarial noise on samples from Figure 6 (Left). (Right) `FAB` Adversarial noise on samples from Figure 6 (Right).

Figure 7 and Figure 8 represent the adversarial noise obtained by an untargeted adversarial attack. Note that some samples in Figure 7 and Figure 8 appear as minor noise (*i.e.* dark sample) since no noise is required to change the class from the true class, it implies that the classifier did not successfully classify the input image.

Experiments conducted in Section 4 reproduce the experimental framework of [21][3]. Figure 9 is the counterpart of Figure 6 in the framework where padding on targeted training data is removed, `FAB` remains the adversarial strategy for both MIA strategy. Comparing the performance of both MIA strategies on both figures, it appears that padding and other preprocessing on train data may favor the success of membership inference attackers, more especially MIA strategy relying on adversarial examples inducing large change in the input (padded) data. In both settings, Sysiphos (Algorithm 1) outperforms [21].

## C.3  Influence of the classifier's pre-trained model

In this section, we assess the influence of Resnet classifier's pretraining and weights on the success of MIA. The PytorchCV public repository[4] provides the weights of Resnet neural networks which are obtained independently from the weights used in Section 4.

Figure 10 is the counterpart of Figure 3 where Resnet's weights are replaced with the weights from PytorchCV. Comparing results from both figures, one can conclude that the target model pretraining has influence on the success of MIAs.

---

[3]https://github.com/ganeshdg95/Leveraging-Adversarial-Examples-to-Quantify-Membership-Information-Leakage

[4]https://pypi.org/project/pytorchcv/

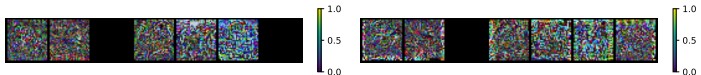

Figure 8: (Left) `AutoAttack` Adversarial noise on samples from Figure 6 (Left). (Right) `AutoAttack` Adversarial noise on samples from Figure 6 (Right).

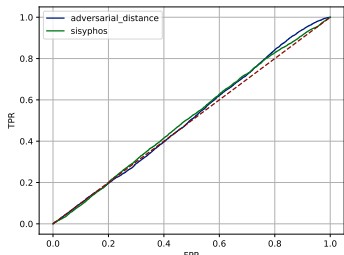

Figure 9: Comparison of ROC curves of MIA scores relying on `FAB` adversarial strategy against Resnet on CIFAR10 without padding in the training set.

# D  Towards a Defense Mechanism

## D.1  Privacy by Design

Most obfuscation mechanisms tend to add noise directly to the samples $X_i$ to reduce the probability of identifying the training data [25]. Alternative means analyze the influence of regularization of neural network [48] or label smoothing [32]. Hereafter we design a mechanism preventing peaks of confidence of the neural network, mitigating MIA related risks. The strategy consists in reducing the vulnerability by adding controlled and voluntary noise to the one-hot encoded labels –counterparts of $Y_i$– while training the deep model. The core idea is the following: *as the model's generalization improves through the training phase, the model should less require simplistic labels.* Algorithm 2 describes the defense mechanism.

---

**Algorithm 2** Defense Mechanism

**Require:** Training dataset $\mathcal{D}_{\text{train}} = \{(X_i, Y_i)\}_{i=1}^n$, class of classifier $\mathcal{G}$, batch size $0 < m \leq n$, a non-decreasing non-negative function $M$ valued in $[0, (K-1)/K]$.
1: Initialize $\widehat{g}_\theta$ with random $\theta \in \Theta$
   Set $L_1 = L_2 = 0$.
   Set $\widetilde{Y}_i = (\mathbb{1}\{j = Y_i\})_{j=1}^K$.
   Set $0 < \zeta < (K-1)/K$.
2: **while** $\theta$ not converged **or** $L_1 \leq L_2$ **do**
3:    Sample $\{(X_1, \widetilde{Y}_1), \ldots, (X_m, \widetilde{Y}_m)\}$ from $\mathcal{D}_{\text{train}}$.
4:    Update $L_1 = L_2$.
5:    Update $L_2 = 1/m \sum_{i=1}^m \ell(\widetilde{Y}_i, \widehat{g}_\theta(X_i))$.
6:    Update $\theta$ by descending $L_2$.
7:    Update $\widetilde{Y}_i = [\mathbb{1}\{j = Y_i\}(1 - \frac{K}{K-1}\zeta) + \frac{\zeta}{K-1}]_{j=1}^K$.
8:    Update $\zeta = M(\zeta)$.
9: **end while**
10: **Return** $\widehat{g}_\theta$.

---

The defense method described above do not change label from sample $X_i$ with some randomization to another label: it provides a mixture of labels in a similar way as label smoothing [71].

**Remark 2 (Random noise injection)** *The label noise may be applied on a random subset of a batch instead of the whole batch. In this way, Appendix D.2 provides an alternative of Algorithm 2 with a randomized noise injection.*

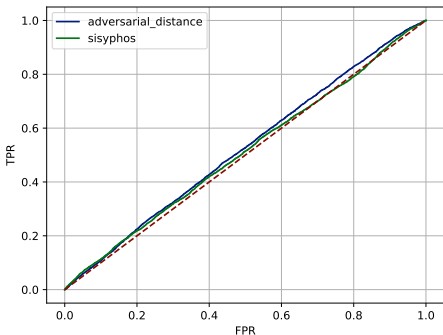

Figure 10: Comparison of ROC curves of MIA scores relying on `FAB` adversarial strategy against Resnet Pytorch.

## D.2 Alternative Defense Mechanism

In Section D.1, we introduce the defense mechanism relying on noisy labels to build a classifier $\widehat{g}_\theta$ hopefully robust to MIA based on adversarial examples. Algorithm 3 provides an alternative version of Algorithm 2 where the voluntary label noise is randomly injected in a subset of the batches of while training. The experimental influence of the randomization of Algorithm 2 is depicted in Section D.3.

---

**Algorithm 3** Randomized SISYPHOS

---

**Require:** Training dataset $\mathcal{D}_{\text{train}} = \{(X_i, Y_i)\}_{i=1}^n$, class of classifier $\mathcal{G}$, batch size $0 < m \leq n$, a non-decreasing non-negative function $M$ valued in $[0, (K-1)/K]$, noise injection probability $p \in ]0, 1[$.
1: Initialize $\widehat{g}_\theta$ with random $\theta \in \Theta$.
   Set $L_1 = L_2 = 0$.
   Set $\widetilde{Y}_i = (\mathbb{1}\{j = Y_i\})_{j=1}^K$.
   Set $0 < \zeta < (K-1)/K$.
2: **while** $\theta$ not converged **or** $L_1 \leq L_2$ **do**
3:     Sample $\{(X_1, \widetilde{Y}_1), \ldots, (X_m, \widetilde{Y}_m)\}$ from $\mathcal{D}_{\text{train}}$.
4:     Update $L_1 = L_2$.
5:     Update $L_2 = 1/m \sum_{i=1}^m \ell(\widetilde{Y}_i, \widehat{g}_\theta(X_i))$.
6:     Update $\theta$ by descending $L_2$.
7:     **for** $i$ **in** $\{1, \ldots, m\}$ **do**
8:         With probability $p$ update $\widetilde{Y}_i$,
$$\widetilde{Y}_i = \left(\mathbb{1}\{j = Y_i\}(1 - \tfrac{K}{K-1}\zeta) + \tfrac{\zeta}{K-1}\right)_{j=1}^K.$$
9:     **end for**
10:    Update $\zeta = M(\zeta)$.
11: **end while**
12: **Return** $\widehat{g}_\theta$.

---

## D.3 Numerical Experiments regarding our Defense Mechanism

In this section, we present numerical experiments run to illustrate the defense mechanism depicted in Section D and its alternative algorithm from Section D.2.

**Dataset.** We work with MNIST dataset [23]: a database of handwritten digits with a training set of 60,000 examples and a test set of 10,000 examples. The digits have been size-normalized and centered in a fixed-size image. The original black and white (bilevel) images from NIST were size normalized to fit in a 20x20 pixel box while preserving their aspect ratio. The resulting images contain gray levels as a result of the anti-aliasing technique used by the normalization algorithm. The images were centered in a 28x28 image by computing the center of mass of the pixels, and translating

the image so as to position this point at the center of the 28x28 field. Each training and test example is assigned to the corresponding handwritten digit between 0 and 9.

**Experimental framework.** The target model is a multi-layer perceptron and the defense mechanisms are Algorithm 2 and Algorithm 3. For both defense mechanisms, the label noise is constant through training (*i.e.* for any $\zeta$, $M(\zeta) = \zeta$) similarly to label smoothing. We train a multi-layer perceptron with one layer containing 100 neurons.

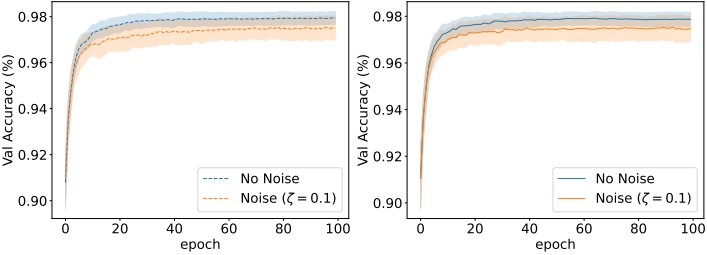

Figure 11: Comparison of $\widehat{g}_\theta$'s accuracies (%) on the validation set during the training phase of $\widehat{g}_\theta$ with (Left) Algorithm 2 and (Right) Algorithm 3 with $p = 0.5$. The amount of label noise for both algorithms is set to $\zeta = 0.1$.

**Results.** Figure 11 provides the evolution of $\widehat{g}_\theta$'s accuracy on the validation set when label noise is injected (with noise level $\zeta = 0.1$). One can observe similar accuracy between the two settings considered. Figure 12 represents the loss values of $\widehat{g}_\theta$ for the validation and train sets through training with noisy labels following Algorithm 2 or Algorithm 3.

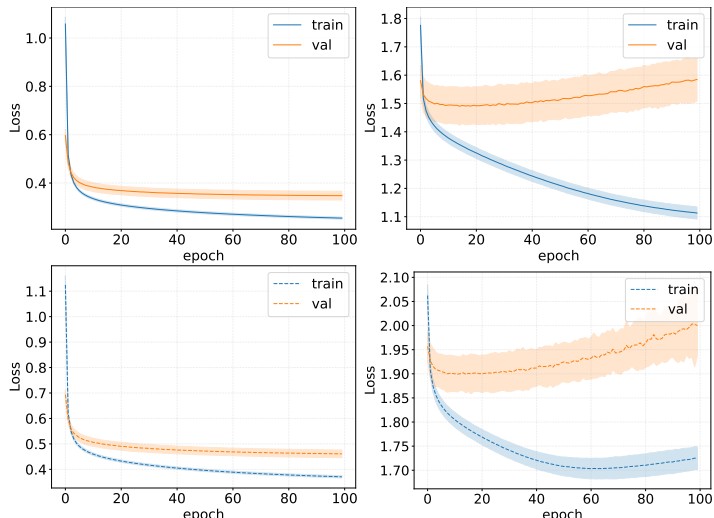

Figure 12: Comparison of $\widehat{g}_\theta$'s loss on the train and validation sets during the training phase of $\widehat{g}_\theta$. The top row represents Algorithm 2 while the bottom row gathers results from Algorithm 3 with $p = 0.5$. The left column corresponds to noise level $\zeta = 0.1$ and the right column when the noise level $\zeta = 0.9$.

Interestingly, one can observe in Figure 12 (Bottom Right) that when large noise ($\zeta = 0.9$) is added to the training labels, two antagonist forces seem to be present: first, the loss minimization focuses on the learning task (*i.e.* predicting the correct class for each sample with the best accuracy possible) and then starts overfitting to the voluntary injected noise in the remaining labels which thus decreases the maximum score granted to the predicted label. In this way, such noise injection could be relevant to better assess early-stopping while training neural networks in addition to some improved robustness to MIAs.

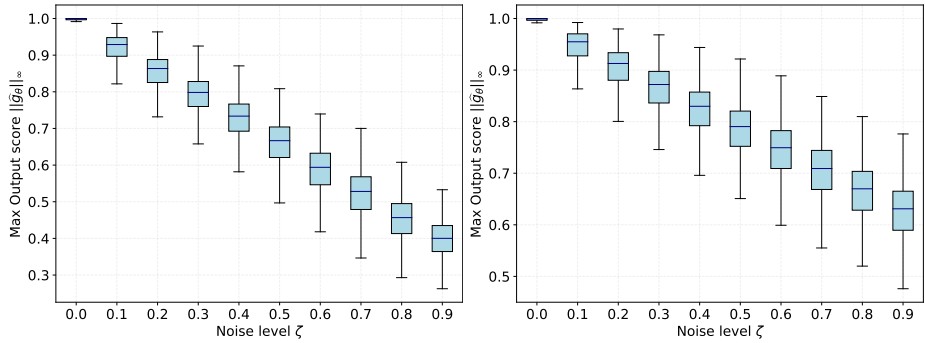

Figure 13: Boxplots of the excess of confidence of $\widehat{g}_\theta$ for varying noise levels.

Figure 13 describes the evolution of the excess of confidence of $\widehat{g}_\theta$ with varying noise levels for (Left) Algorithm 2 and (Right) Algorithm 3. We can see that as more noise is injected in the labels, the excess confidence of $\widehat{g}_\theta$ decreases, thus being more robust to MIAs based on adversarial examples.

## E  Broader MIA Experiments

The MIA framework in this paper targets to predict if some sample $(X_i, Y_i) \in \mathcal{D}_{\text{test}}$ belongs to from the training set $\mathcal{D}_{\text{train}}$, under the assumption that $\mathcal{D}_{\text{test}} \cap \mathcal{D}_{\text{train}} \neq \emptyset$. In practice, access to the exact samples $(X_i, Y_i)$ is unlikely. In this section, we want to assess if classifier's excess of confidence can still be noticed when one has solely access to $(\widetilde{X}_i, Y_i)$ with $\widetilde{X}_i$ remaining close to $X_i$. The experiment detailed in this section suggests that the adversarial attack designed in this paper may be generalized to more complex and real world settings.

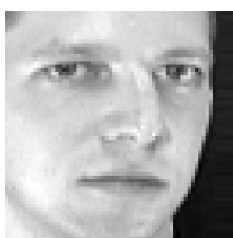 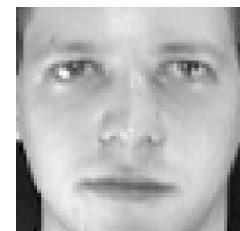 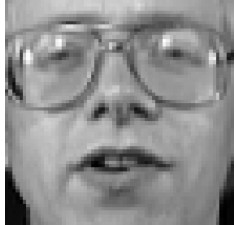

Figure 14: Samples from Olivetti faces dataset.

**Dataset.** The Olivetti dataset contains a set of face images taken between April 1992 and April 1994 at AT&T Laboratories Cambridge. There are ten different images $\{X_i^j\}_{i=1}^{10}$ of each distinct subject $j$ among the 40 subjects. The label $Y_i$ associated to a sample $X_i$ consists in assessing if the person in $X_i$ wears glasses (i.e. $Y_i = 1$ if and only if the input image $X_i$ contains glasses). For some subjects, the images were taken at different times, varying the lighting, facial expressions (open / closed eyes, smiling / not smiling) and facial details (glasses / no glasses). All the images were taken against a dark homogeneous background with the subjects in an upright, frontal position (with tolerance for some side movement). Figure 14 illustrates samples $X_1^j, X_2^j$ (Left, Center) of subject $j$ and $X_1^{j'}$ (Right) of subject $j'$. Subject $j$ wears no glasses in both presented images while subject $j'$ wears glasses in the presented image.

**Experimental framework.** Let $\mathcal{D}_{\text{train}} = \{(X_i^j, Y_i)\}_{i<10}^{j \leq 39}$ denote the training set used to build a classifier $\widehat{g}_\theta$. Let $\mathcal{D}_{\text{test}} = \{(X_i^j, Y_i)\}_{i=10}^{j \leq 39} \cup \{(X_i, Y_i)\}_{i \leq 10}^{j=40}$ contain the remaining samples. By construction, there is no sample $X_i$ belonging to the intersection between $\mathcal{D}_{\text{train}}$ and $\mathcal{D}_{\text{test}}$, although some subjects belong to both $\mathcal{D}_{\text{train}}$ and $\mathcal{D}_{\text{test}}$. $\mathcal{D}_{\text{test}}$ contains 10 images of subject 40 unseen in the training set. We compare the maximum scores of $\widehat{g}_\theta$ for new samples of subjects seen in the training data and new samples of subject unseen in the training phase.

**Results.** Boxplots in Figure 15 report the max score of $\widehat{g}_\theta$ in two subsets of $\mathcal{D}_{\text{test}}$. The mean of the scores is denoted in orange. The median of the scores is denoted in dashed green. Figure 15 essentially suggests that exploiting the excess of confidence of a classifier to retrieve training data is possible even only with access to modified inputs of the training set. The classifier $\widehat{g}_\theta$ shows larger confidence on its prediction when subjects have already been encountered in the training phase. Therefore, one may generalize MIA strategies with adversarial examples relying on functional in the output space of $\widehat{g}_\theta$ to real world settings.

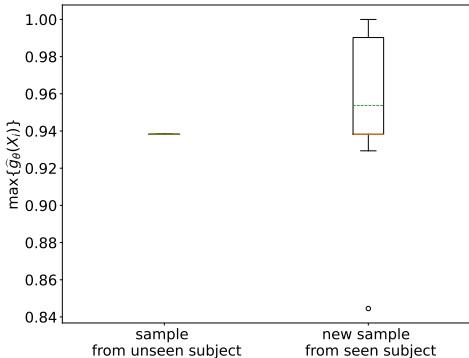

Figure 15: (Left) Boxplots of the max score $\widehat{g}_\theta$ on samples $X_i^{40}$s from subject $40$ from the test set $\mathcal{D}_{\text{test}}$. (Right) Boxplots of the max score $\widehat{g}_\theta$ on the remainder of samples in the test set $\mathcal{D}_{\text{test}}$ with subjects seen in $\mathcal{D}_{\text{train}}$.

