# OpenReview forum: "Membership Inference Attacks via Adversarial Examples"
_NeurIPS.cc/2022/Workshop/TSRML — TSRML2022_

### Official Review · Reviewer_KS5D · 2022-10-21
**Well-motivated though empirical results slightly weak**

**Overall Recommendation:** I recommend acceptance of this work.
**Overall Rating:** 8

**Summary:**

This paper studies a new membership inference attack, which is the most common privacy attack, via the adversarial examples. Consequently a new defense is developed.

**Strengths:**

This paper is well-written, with enlightening observation of "peaks of confidence" and good argument about adversarial path. The resulting algorithms are clear and moderately effective. Overall the contribution is solid.

**Weaknesses:**

The empirical performance is not strong and computational cost may be concerning. Also, given that the motivation is empirical, it would be desirable to look at MNIST instead of synthetic dataset.

**Review Confidence:**

4: The reviewer is confident but not absolutely certain that the evaluation is correct

---

### Decision · Program_Chairs · 2022-10-23

Accept